# Understanding Regularized Spectral Clustering via Graph Conductance

**Yilin Zhang**
Department of Statistics
University of Wisconsin-Madison
Madison, WI 53706
yilin.zhang@wisc.edu

**Karl Rohe**
Department of Statistics
University of Wisconsin-Madison
Madison, WI 53706
karl.rohe@wisc.edu

## Abstract

This paper uses the relationship between graph conductance and spectral clustering to study (i) the failures of spectral clustering and (ii) the benefits of regularization. The explanation is simple. Sparse and stochastic graphs create several "dangling sets", or small trees that are connected to the core of the graph by only one edge. Graph conductance is sensitive to these noisy dangling sets and spectral clustering inherits this sensitivity. The second part of the paper starts from a previously proposed form of regularized spectral clustering and shows that it is related to the graph conductance on a "regularized graph". When graph conductance is computed on the regularized graph, we call it CoreCut. Based upon previous arguments that relate graph conductance to spectral clustering (e.g. Cheeger inequality), minimizing CoreCut relaxes to regularized spectral clustering. Simple inspection of CoreCut reveals why it is less sensitive to dangling sets. Together, these results show that unbalanced partitions from spectral clustering can be understood as overfitting to noise in the periphery of a sparse and stochastic graph. Regularization fixes this overfitting. In addition to this statistical benefit, these results also demonstrate how regularization can improve the computational speed of spectral clustering. We provide simulations and data examples to illustrate these results.

## 1 Introduction

Spectral clustering partitions the nodes of a graph into groups based upon the eigenvectors of the graph Laplacian [19, 20]. Despite the claims of spectral clustering being "popular", in applied research using graph data, spectral clustering (without regularization) often returns a partition of the nodes that is uninteresting, typically finding a large cluster that contains most of the data and many smaller clusters, each with only a few nodes. These applications involve brain graphs [2] and social networks from Facebook [21] and Twitter [22]. One key motivation for spectral clustering is that it relaxes a discrete optimization problem of minimizing graph conductance. Previous research has shown that across a wide range of social and information networks, the clusters with the smallest graph conductance are often rather small [15]. Figure 1 illustrates the leading singular vectors on a communication network from Facebook during the 2012 French presidential election [21]. The singular vectors localize on a few nodes, which leads to a highly unbalanced partition.

[1] proposed regularized spectral clustering which adds a weak edge on every pair of nodes with edge weight $\tau/N$, where $N$ is the number of nodes in the network and $\tau$ is a tuning parameter. [5] proposed a related technique. Figure 1 illustrates how regularization changes the leading singular vectors in the Facebook example. The singular vectors are more spread across nodes.

Many empirical networks have a core-periphery structure, where nodes in the core of the graph are more densely connected and nodes in the periphery are sparsely connected [3]. In Figure 1,

regularized spectral clustering leads to a "deeper cut" into the core of the graph. In this application, regularization helps spectral clustering provide a more balanced partition, revealing a more salient political division.

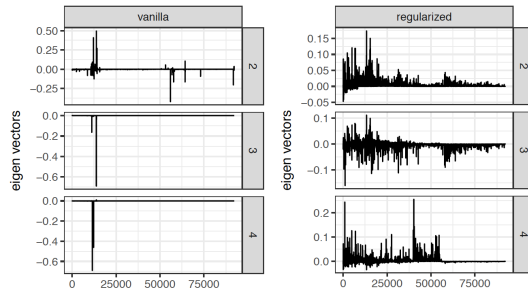

Figure 1: This figure shows the leading singular vectors of the communication network. In the left panel, the singular vectors from vanilla spectral clustering are localized on a few nodes. In the right panel, the singular vectors from regularized spectral clustering provide a more balanced partition.

Previous research has studied how regularization improves the spectral convergence of the graph Laplacian [17, 9, 11]. This paper aims to provide an alternative interpretation of regularization by relating it to graph conductance. We call spectral clustering without regularization `Vanilla-SC` and with edge-wise regularization `Regularized-SC` [1].

This paper demonstrates (1) what makes `Vanilla-SC` fail and (2) how `Regularized-SC` fixes that problem. One key motivation for `Vanilla-SC` is that it relaxes a discrete optimization problem of minimizing graph conductance [7]. Yet, this graph conductance problem is fragile to small cuts in the graph. The fundamental fragility of graph conductance that is studied in this paper comes from the type of subgraph illustrated in Figure 2 and defined here.

**Definition 1.1.** In an unweighted graph $G = (V, E)$, subset $S \subset V$ is $g$-**dangling** if and only if the following conditions hold.

- $S$ contains exactly $g$ nodes.
- There are exactly $g - 1$ edges within $S$ and they do not form any cycles (i.e. the node induced subgraph from $S$ is a tree).
- There is exactly one edge between nodes in $S$ and nodes in $S^c$.

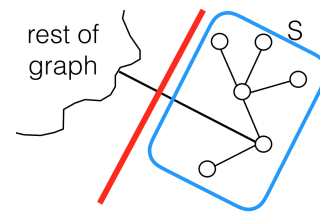

Figure 2: 6-dangling set.

The argument in this paper is structured as follows:

1) A $g$-dangling set has a small graph conductance, approximately $(2g)^{-1}$ (Section 3.2).

2) For any fixed $g$, graphs sampled from a sparse inhomogeneous model with $N$ nodes have $\Theta(N)$ $g$-dangling sets in expectation (Theorem 3.4). As such, $g$-dangling sets are created as an artifact of the sparse and stochastic noise.

3) This makes $\Theta(N)$ eigenvalues in the normalized graph Laplacian which have an average value less than $(g - 1)^{-1}$ (Theorem 3.5) and reveal only noise. These small eigenvalues are so numerous that they conceal good cuts to the core of the graph.

4) $\Theta(N)$ eigenvalues smaller than $1/g$ also make the eigengap exceptionally small. This slows down the numerical convergence for computing the eigenvectors and values.

5) `CoreCut`, which is graph conductance on the regularized graph, does not assign a small value to small sets of nodes. This prevents all of the statistical and computational consequences listed above for $g$-dangling sets and any other small noisy subgraphs that have a small conductance. `Regularized-SC` inherits the advantages of `CoreCut`.

The penultimate section evaluates the overfitting of spectral clustering in an experiment with several empirical graphs from SNAP [14]. This experiment randomly divides the edges into training set

and test set, then runs spectral clustering using the training edges and with the resulting partition, compares "training edge conductance" to "testing edge conductance." This shows that `Vanilla-SC` overfits and `Regularized-SC` does not. Moreover, `Vanilla-SC` tends to identify highly unbalanced partitions, while `Regularized-SC` provides a balanced partition.

The paper concludes with a discussion which illustrates how these results might help inform the construction of neural architectures for a generalization of Convolutional Neural Networks to cases where the input data has an estimated dependence structure that is represented as a graph [12, 4, 10, 16].

## 2  Notation

**Graph notation**    The graph or network $G = (V, E)$ consists of node set $V = \{1, \ldots, N\}$ and edge set $E = \{(i, j) : i \text{ and } j \text{ connect with each other}\}$. For a weighted graph, the edge weight $w_{ij}$ can take any non-negative value for $(i, j) \in E$ and define $w_{ij} = 0$ if $(i, j) \notin E$. For an unweighted graph, define the edge weight $w_{ij} = 1$ if $(i, j) \in E$ and $w_{ij} = 0$ otherwise. For each node $i$, we denote its degree as $d_i = \sum_j w_{ij}$. Given $S \subset V$, the node induced subgraph of $S$ in $G$ is a graph with vertex set $S$ and includes every edge whose end point are both in $S$, i.e. its edge set is $\{(i, j) \in E : i, j \in S\}$.

**Graph cut notation**    For any subset $S \subset V$, we denote $|S| =$ number of nodes in $S$, and its volume in graph $G$ as $vol(S, G) = \sum_{i \in S} d_i$. Note that any non-empty subset $S \subsetneq V$ forms a partition of $V$ with its complement $S^c$. We denote the cut for such partition on graph $G$ as

$$cut(S, G) = \sum_{i \in S, j \in S^c} w_{ij},$$

and denote the graph conductance of any subset $S \subset V$ with $vol(S, G) \leq vol(S^c, G)$ as

$$\phi(S, G) = \frac{cut(S, G)}{vol(S, G)}.$$

Without loss of generality, we focus on non-empty subsets $S \subsetneq V$ with $vol(S, G) \leq vol(S^c, G)$.

**Notation for `Vanilla-SC` and `Regularized-SC`**    We denote the adjacency matrix $A \in \mathbb{R}^{N \times N}$ with $A_{ij} = w_{ij}$, and the degree matrix $D \in \mathbb{R}^{N \times N}$ with $D_{ii} = d_i$ and $D_{ij} = 0$ for $i \neq j$. The normalized graph Laplacian matrix is

$$L = I - D^{-1/2} A D^{-1/2},$$

with eigenvalues $0 = \lambda_1 \leq \lambda_2 \leq \ldots \lambda_N \leq 2$ (here and elsewhere, "leading" refers to the smallest eigenvalues). Let $v_1, \ldots, v_N : V \to \mathbb{R}$ represent the eigenvectors/eigenfunctions for $L$ corresponding to eigenvalues $\lambda_1, \ldots, \lambda_N$.

There is a broad class of spectral clustering algorithms which represent each node $i$ in $\mathbb{R}^K$ with $(v_1(i), \ldots, v_K(i))$ and cluster the nodes by clustering their representations in $\mathbb{R}^K$ with some algorithm. For simplicity, this paper focuses on the setting of $K = 2$ and only uses $v_2$. We refer to `Vanilla-SC` the algorithm which returns the set $S_i$ which solves

$$\min_i \phi(S_i, G), \text{ where } S_i = \{j : v_2(j) \geq v_2(i)\}. \tag{2.1}$$

This construction of a partition appears in both [19] and in the proof of Cheeger inequality [6, 7], which says that

$$\textbf{Cheeger inequality:} \quad \frac{h_G^2}{2} \leq \lambda_2 \leq 2h_G, \text{ where } h_G = \min_S \phi(S, G).$$

Edge-wise regularization [1] adds $\tau/N$ to every element of the adjacency matrix, where $\tau > 0$ is a tuning parameter. It replaces $A$ by matrix $A_\tau \in \mathbb{R}^{N \times N}$, where $[A_\tau]_{ij} = A_{ij} + \tau/N$ and the node degree matrix $D$ by $D_\tau$, which is computed with the row sums of $A_\tau$ (instead of the row sums of $A$) to get $[D_\tau]_{ii} = D_{ii} + \tau$. We define $G_\tau$ to be the weighted graph with adjacency matrix $A_\tau$ as defined above. `Regularized-SC` partitions the graph using the $K$ leading eigenvectors of $L_\tau = I - D_\tau^{-1/2} A_\tau D_\tau^{-1/2}$, which we represent by $v_1^\tau, \ldots, v_K^\tau : V \to \mathbb{R}$. Similarly, we only use $v_2^\tau$ when $K = 2$. We refer to `Regularized-SC` the algorithm which returns the set $S_i$ which solves

$$\min_i \phi(S_i, G_\tau), \text{ where } S_i = \{j : v_2^\tau(j) \geq v_2^\tau(i)\}.$$

# 3 `Vanilla-SC` and the periphery of sparse and stochastic graphs

For notational simplicity, this section only considers unweighted graphs.

## 3.1 Dangling sets have small graph conductance.

The following fact follows from the definition of a $g$-dangling set.

**Fact 3.1.** *If $S$ is a $g$-dangling set, then its graph conductance is $\phi(S) = (2g - 1)^{-1}$.*

To interpret the scale of this graph conductance, imagine that a graph is generated from a Stochastic Blockmodel with two equal-size blocks, where any two nodes from the same block connect with probability $p$ and two nodes from different blocks connect with probability $q$ [8]. Then, the graph conductance of one of the blocks is $q/(p + q)$ (up to random fluctuations). If there is a $g$-dangling set with $g > p/(2q) + 1$, then the $g$-dangling set will have a smaller graph conductance than the block.

## 3.2 There are many dangling sets in sparse and stochastic social networks.

We consider random graphs sampled from the following model which generalizes Stochastic Blockmodels. Its key assumption is that edges are independent.

**Definition 3.2.** A graph is generated from an **inhomogeneous random graph model** if the vertex set contains $N$ nodes and all edges are independent. That is, for any two nodes $i, j \in V$, $i$ connects to $j$ with some probability $p_{ij}$ and this event is independent of the formation of any other edges. We only consider undirected graphs with no self-loops.

**Definition 3.3.** Node $i$ is a **peripheral node** in an inhomogeneous random graph with $N$ nodes if there exist some constant $b > 0$, such that $p_{ij} < b/N$ for all other nodes $j$, where we allow $N \to \infty$.

For example, an Erdös-Rényi graph is an inhomogeneous random graph. If the Erdös-Rényi edge probability is specified by $p = \lambda/N$ for some fixed $\lambda > 0$, then all nodes are peripheral. As another example, a common assumption in the statistical literature on Stochastic Blockmodels is that the minimum expected degree grows faster than $\log N$. Under this assumption, there are no peripheral nodes in the graph. That $\log N$ assumption is perhaps controversial because empirical graphs often have many low-degree nodes.

**Theorem 3.4.** *Suppose an inhomogeneous random graph model such that for some $\epsilon > 0$, $p_{ij} > (1 + \epsilon)/N$ for all nodes $i, j$. If that model contains a non-vanishing fraction of peripheral nodes $V_p \subset V$, such that $|V_p| > \eta N$ for some $\eta > 0$, then the expected number of distinct g-dangling sets in the sampled graph grows proportionally to $N$.*

Theorem 3.4 studies graphs sampled from an inhomogeneous random graph model with a non-vanishing fraction of peripheral nodes. Throughout the paper, we refer to these graphs more simply as graphs with a sparse and stochastic periphery and, in fact, the proof of Theorem 3.4 only relies on the randomness of the edges in the periphery, i.e. the edges that have an end point in $V_p$. The proof does not rely on the distribution of the node-induced subgraph of the "core" $V_p^c$. Combined with Fact 3.1, Theorem 3.4 shows that graphs with a sparse and stochastic periphery generate an abundance of $g$-dangling sets, which creates an abundance of cuts with small conductance, but might only reveal noise. [15] also shows by real datasets that there is a substantial fraction of nodes that barely connect to the rest of graph, especially 1-whiskers, which is a generalized version of g-dangling sets.

**Theorem 3.5.** *If a graph contains $Q$ g-dangling sets, and the rest of the graph has volume at least $4g^2$, then there are at least $Q/2$ eigenvalues that is smaller than $(g - 1)^{-1}$.*

Theorem 3.5 shows that every two dangling sets lead to a small eigenvalue. Due to the abundance of $g$-dangling sets (Theorem 3.4), there are many small eigenvalues and their corresponding eigenvalues are localized on a small set of nodes. This explains what we see in the data example in Figure 1. Each of these many eigenvectors is costly to compute (due to the small eigengaps) and then one needs to decide which are localized (which requires another tuning).

# 4 `CoreCut` ignores small cuts and relaxes to `Regularized-SC`.

Similar to the graph conductance $\phi(\cdot, G)$ which relaxes to `Vanilla-SC` [7, 19, 20], we introduce a new graph conductance `CoreCut` which relaxes to `Regularized-SC`. The following sketch illustrates

the relations. This section compares $\phi(\cdot, G)$ and `CoreCut`. For ease of exposition, we continue to focus our attention on partitioning into two sets.

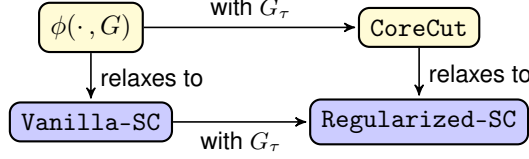

**Definition 4.1.** Given a subset $S \subset V$ with $vol(S, G_\tau) \le vol(S^c, G_\tau)$, we define its `CoreCut` as

$$\texttt{CoreCut}_\tau(S) = \frac{cut(S, G) + \frac{\tau}{N}|S||S^c|}{vol(S, G) + \tau|S|}.$$

**Fact 4.2.** *For any $S \subset V$ with $vol(S, G_\tau) \le vol(S^c, G_\tau)$, it follows that $\texttt{CoreCut}_\tau(S) = \phi(S, G_\tau)$.*

With Fact 4.2, we can apply Cheeger inequality to $G_\tau$ in order to relate the optimum `CoreCut` to the second eigenvalue of $L_\tau$, which we denote by $\lambda_2(L_\tau)$.

$$\frac{h_\tau^2}{2} \le \lambda_2(L_\tau) \le 2h_\tau \ \text{ where } h_\tau = \min_S \texttt{CoreCut}_\tau(S).$$

The fundamental property of `CoreCut` is that the regularizer $\tau$ has larger effect on smaller sets. For example in Figure 3a, $S_{\epsilon_i}$'s ($i = 1, ..., 5$) are small peripheral sets and $S_1$, $S_2$ are core sets, each with roughly half of all nodes. From Figure 3, all five peripheral sets have smaller $\phi(\cdot, G)$ than the two core sets. Minimizing $\phi(\cdot, G)$ tends to cut the periphery rather than cutting the core. By regularizing with $\tau = 2$, the `CoreCut` of all five peripheral sets increases significantly from $\phi(\cdot, G)$, while `CoreCut` of the two core sets remain similar to their $\phi(\cdot, G)$. In the end, `CoreCut` will cut the core of the graph because all five peripheral sets have larger `CoreCut` than the two core sets $S_1, S_2$.

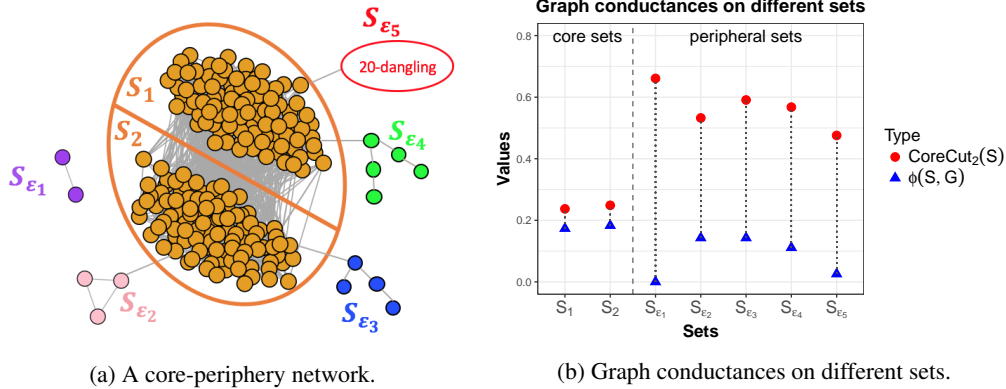

(a) A core-periphery network.  (b) Graph conductances on different sets.

Figure 3: Figure (b) shows the `CoreCut` with $\tau = 2$, and $\phi(\cdot, G)$ on different sets in the core-periphery network in Figure (a). `CoreCut` is very close to $\phi(\cdot, G)$ on the core sets $S_1$ and $S_2$. But on the peripheral sets, $\phi(\cdot, G)$ assigns small values, while `CoreCut` assigns much larger values. Minimizing $\phi(\cdot, G)$ will yield a peripheral set, while minimizing `CoreCut` will cut the core of the graph.

`CoreCut` will succeed if $\tau$ overwhelms the peripheral sets, but is negligible to core sets. Corollary 4.7 below makes this intuition precise. It requires the following assumptions, where you should imagine $S_\epsilon$ to be a peripheral cut and $S$ to be a cut to the core of the graph that we wish to detect.

We define the mean degree for any subset $S' \subset V$ on $G$ as $\bar{d}(S', G) = vol(S', G)/|S'|$.

**Assumption 4.3.** *For a graph $G = (V, E)$ and subsets $S_\epsilon \subset V$ and $S \subset V$, there exists $\epsilon, \alpha > 0$, such that*

1. $|S_\epsilon| < \epsilon|V|$ *and* $vol(S_\epsilon, G) < \epsilon vol(V, G)$,

2. $\bar{d}(S_\epsilon, G) < \frac{1-\epsilon}{2(1+\alpha)}\bar{d}(S, G)$,

3. $\phi(S, G) < \frac{\alpha(1-\epsilon)}{1+\alpha}$.

**Remark 4.4.** Assumption 1 indicates that the peripheral set $S_\epsilon$ is a very small part of $G$ in terms of number of nodes and number of edges. Assumption 2 requires $S$ to be reasonably dense. Assumption 3 requires $S$ and $S^c$ to form a good partition.

**Proposition 4.5.** *Given graph* $G = (V, E)$, *for any set* $S_\epsilon \subset V$ *satisfying Assumption 1, for some constant* $\alpha > 0$, *if we choose* $\tau$ *such that* $\tau \geq \alpha\bar{d}(S_\epsilon)$, *then*

$$\mathtt{CoreCut}_\tau(S_\epsilon) > \frac{\alpha(1-\epsilon)}{1+\alpha}.$$

Proposition 4.5 shows that `CoreCut` of a peripheral set is lower bounded away from zero.

**Proposition 4.6.** *Given graph* $G = (V, E)$, *for any set* $S \subset V$, *for some constant* $\delta > 0$, *if we choose* $\tau \leq \delta\bar{d}(S, G)$, *then*

$$\mathtt{CoreCut}_\tau(S) < \phi(S, G) + \delta.$$

When $S$ is reasonably large, $\tau$ can be chosen such that $\delta$ is small. Proposition 4.6 shows that with $\tau$ not being too large, the `CoreCut` of a reasonably large set is close to $\phi(\cdot, G)$.

Corollary 4.7 follows directly from Proposition 4.5 and 4.6.

**Corollary 4.7.** *Given graph* $G = (V, E)$, *for any subsets* $S_\epsilon, S \subset V$ *satisfying the three assumptions in Assumption 4.3, if we choose* $\tau$ *such that*

$$\alpha\bar{d}(S_\epsilon, G) \leq \tau \leq \delta\bar{d}(S, G),$$

*where* $\delta = \alpha(1 - \epsilon)/(1 + \alpha) - \phi(S, G)$, *then*

$$\mathtt{CoreCut}_\tau(S) < \mathtt{CoreCut}_\tau(S_\epsilon).$$

Corollary 4.7 indicates the lower bound and upper bound of $\tau$ for `CoreCut` to ignore a cut to the periphery and prefer a cut to the core. These bounds on $\tau$ lead to a deeper understanding of `CoreCut`. However, they are difficult to implement in practice.

## 5 Real data examples

This section provides real data examples to show three things. First, `Regularized-SC` finds a more balanced partition. Second, `Vanilla-SC` is prone to "catastrophic overfitting". Third, computing the second eigenvector of $L_\tau$ takes less time than computing the second eigenvector of $L$. This section studies 37 example networks from http://snap.stanford.edu/data [14]. These networks are selected to be relatively easy to interpret and handle. The largest graph used is wiki-talk and has only 2,388,953 nodes in the largest component. The complete list of graphs used is given below. Before computing anything, directed edges are symmetrized and nodes not connected to the largest connected component are removed. Throughout all simulations, the regularization parameter $\tau$ is set to be the average degree of the graph. This is not optimized, but is instead a previously proposed heuristic [17]. As defined in Section 2 Equation 2.1, the partitions are constructed by scanning through the second eigenvector. Even though we argue that regularized approaches are trying to minimize `CoreCut`, every notion of conductance in this section is computed on the unregularized graph $G$, including the scanning through the second eigenvector. All eigen-computations are performed with `rARPACK` [13, 18].

In this simulation, half of the edges are removed from the graph and placed into a "testing-set". Refer to the remaining edges as the "training-edges". On the training-edges, the largest connected component is again identified. Based upon that subset of the training-edges, the spectral partitions are formed.

Each figure in this section corresponds to a different summary value (balance, training conductance, testing conductance, and running time). In all figures, each point corresponds to a single network.

The $x$-axis corresponds to the summary value for `Regularized-SC` and the $y$-axis corresponds to the summary value for `Vanilla-SC`. Each figure includes a black line, which is the line $x = y$. All plots are on the log-log scale. The size of each point corresponds to the number of nodes in the graph.

In Figure 4, the summary value is the number of nodes in the smaller partition set. Notice that the scales of the axes are entirely different. `Vanilla-SC` tends to identify sets with 100s of nodes or smaller. However, regularizing tends increase the size of the sets into the 1000s.

In Figure 5a, the summary value is the conductance computed on the training-edges. Because this is the quantity that `Vanilla-SC` approximates, it is not surprising that it finds partitions with a smaller conductance. However, Figure 5b shows that if the conductance is computed using only edges in the testing-set, then sometimes the vanilla sets have no internal edges ($\phi(\cdot, G) = 1$). We refer to this as catastrophic overfitting.

In these simulations (and others), we find that the partitions produced by both forms of regularization [1] and [5] are exactly equivalent. We find it easier to implement fast code for [5] and moreover, our implementations of it run faster. Implementing [1] to take advantage of the sparsity in the graph requires defining a function which quickly multiplies a vector $x$ by $L_\tau$. This can be done via $L_\tau x = x - D_\tau^{-1/2} A D_\tau^{-1/2} x - \tau/N \mathbf{1}(\mathbf{1}^T x)$, where $\mathbf{1}$ is a vector of 1's. However, with a user defined matrix multiplication, the eigensolver in `rARPACK` runs slightly slower. Because the regularized form from [5] simply defines $L_\tau = I - D_\tau^{-1/2} A D_\tau^{-1/2}$, it can use the same eigensolver as `Vanilla-SC` and, as such, the running times are more comparable. Figure 6 uses this definition of `Regularized-SC`. Running times are from `rARPACK` computing two eigenvectors of $D_\tau^{-1/2} A D_\tau^{-1/2}$ and $D^{-1/2} A D^{-1/2}$ using the default settings. A line of regression is added to Figure 6. The slope of this line is roughly 1.01 and its intercept is roughly 0.83.

The list of SNAP networks is given here: amazon0302, amazon0312, amazon0505, amazon0601, ca-AstroPh, ca-CondMat, ca-GrQc, ca-HepPh, ca-HepTh, cit-HepPh, cit-HepTh, com-amazon.ungraph, com-youtube.ungraph, email-EuAll, email-Eu-core, facebook-combined, p2p-Gnutella04, p2p-Gnutella05, p2p-Gnutella06, p2p-Gnutella08, p2p-Gnutella09, p2p-Gnutella24, p2p-Gnutella25, p2p-Gnutella30, p2p-Gnutella31, roadNet-CA, roadNet-PA, roadNet-TX, soc-Epinions1, soc-Slashdot0811, soc-Slashdot0902, twitter-combined, web-Google, web-NotreDame, web-Stanford, wiki-Talk, wiki-Vote.

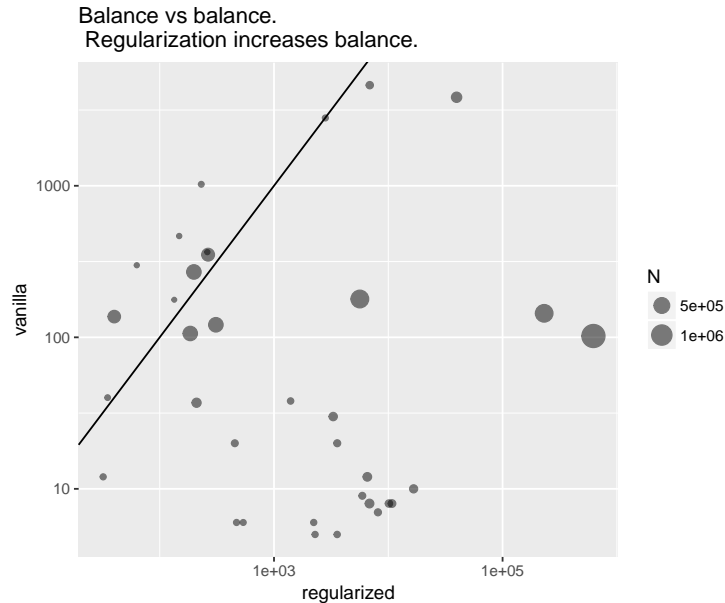

Figure 4: `Regularized-SC` identifies clusters that are more balanced. That is, the smallest set in the partition has more nodes.

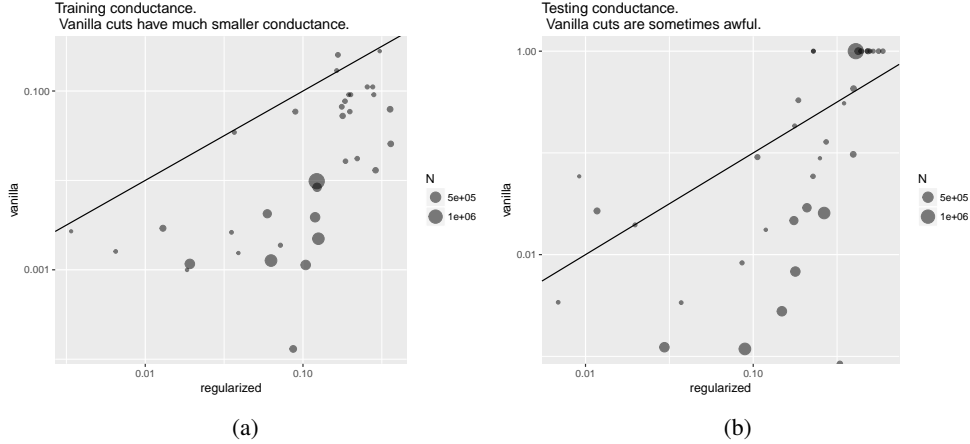

Figure 5: `Vanilla-SC` finds cuts with a smaller conductance. However, on the testing edges, it can have a catastrophic failure, where there are no internal edges to the smallest set. This corresponds to $\phi(\cdot, G) = 1$.

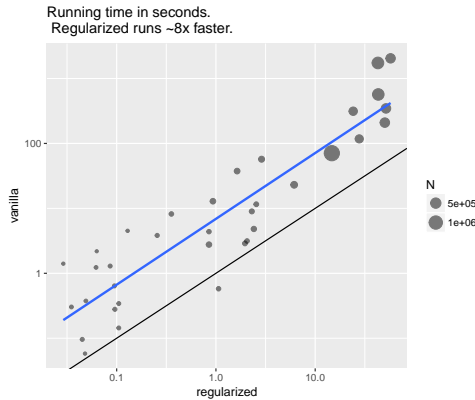

Figure 6: The line of regression suggests that `Regularized-SC` runs roughly eight times faster than `Vanilla-SC` in `rARPACK` [18].

## 6   Discussion

The results in this paper provide a refined understanding of how regularized spectral clustering prevents overfitting. This paper suggests that spectral clustering overfits to $g$-dangling sets (and, perhaps, other small sets) because they occur as noise in sparse and stochastic graphs and they have a very small cost function $\phi$. Regularized spectral clustering optimizes a relaxation of `CoreCut` (a cost function very much related to $\phi$) that assigns a higher cost to small sets like $g$-dangling sets. As such, when a graph is sparse and stochastic, the patterns identified by regularized spectral clustering are more likely to persist in another sample of the graph from the same distribution.

Such overfitting on peripheries may also happen in many other machine learning methods with graph data. There has been an interest in generalizing Convolutional Neural Networks beyond images, to more general graph dependence structures. In these settings, the architecture of the first layer should identify a localized region of the graph [12, 4, 10, 16]. While spectral approaches have been proposed, our results herein suggest potential benefits from regularization.

## Acknowledgements

The authors gratefully acknowledge support from NSF grant DMS-1612456 and ARO grant W911NF-15-1-0423. We thank Yeganeh Ali Mohammadi and Mobin YahyazadehJeloudar for their helpful comments.

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
