[Supplementary Material · supplementary.pdf]

# Supplementary Materials for Understanding Regularized Spectral Clustering via Graph Conductance

**Yilin Zhang**
Department of Statistics
University of Wisconsin-Madison
Madison, WI 53706
yilin.zhang@wisc.edu

**Karl Rohe**
Department of Statistics
University of Wisconsin-Madison
Madison, WI 53706
karl.rohe@wisc.edu

This supplementary consists of four parts. Section 1 provides proof for Theorem 3.4. Section 2 provides proof for Theorem 3.5. Section 3 includes proof for Corollary 4.7. Section 4 contains real data examples.

## 1 Proof of Theorem 3.4

The number of *non-overlapping* $g$-dangling sets is upper bounded by $N$. Actually, by Lemma 1.2 below, it is upper bounded by $N/g$.

To provide a lower bound on the number of *non-overlapping* $g$-dangling sets, we add a fourth requirement to the $g$-dangling definition and call this $g$-dangling. The first three conditions are identical to $g$-dangling. The fourth condition will be satisfied by ensuring that the external edge connects to the largest connected component of the graph. Note that we use this additional fourth condition in the proof to ensure we are counting non-overlapping dangling sets (Lemma 1.2). Since we only provide the lower bound in Theorem 3.4 and this condition only reduces the number of dangling sets, the same result applies for the Definition 1.1 in paper.

**Definition 1.1.** In a graph $G = (V, E)$, for any subset $S \subset V$, $S$ is $g$-dangling if and only if the following conditions hold.

- $S$ contains exactly $g$ nodes.

- There are exactly $g - 1$ edges within $S$ that do not form any cycles (i.e. the node induced subgraph from $S$ is a tree).

- There is exactly one edge between nodes in $S$ and nodes in $S^c$.

- $S$ is part of a connected component in $G$ that has at least $10g$ nodes.

**Lemma 1.2.** *If $S$ is $g$-dangling and $i \in S$, then there is no other $g$-dangling set that contains $i$.*

The proof Lemma 1.2 is at the end of the proof for Theorem 3.4. The proof of Theorem 3.4 is given below.

*Proof.* Due to Lemma 1.2, the number of $g$-dangling sets is a lower bound for the number of non-overlapping $g$-dangling sets. Therefore, it is enough to prove that the expected number of $g$-dangling sets is lower bounded by $\epsilon N$ for some $\epsilon > 0$. Denote the number of $g$-dangling sets as

$$D_g = \sum_{S:|S|=g} \mathbb{1}\{S \text{ is } g\text{-dangling}\},$$

where $\mathbb{1}$ is an indicator function.

In what follows $\epsilon_i > 0$ is a positive constant for $i = 1, \ldots$ that only requires $N$ to be large enough. The $\epsilon_i$'s could have dependence on $g$, which we consider fixed.

In the proof below, we decompose $\mathbb{P}\{S \text{ is } g\text{-dangling}\}$ into a product of several probabilities. First, decompose $\{S \text{ is } g\text{-dangling}\}$ into $T(S) \cap O(S) \cap C(S)$, where

$$
\begin{aligned}
T(S) &= \{\text{the node induced subgraph from } S \text{ is a tree}\} \\
O(S) &= \{\text{the nodes in } S \text{ have one external connection}\} \\
C(S) &= \{S \text{ is part of a connected component that contains at least } 10g \text{ nodes}\}
\end{aligned}
$$

The first two events are independent because it is an inhomogeneous random graph. Then, $T(S)$ further decomposes. For that decomposition, denote $|E(S)|$ as the number of edges in the node induced subgraph from $S$.

$$
\begin{aligned}
\mathbb{E}D_g &\geq \sum_{S \subset |V_p| : |S| = g} \mathbb{P}\{S \text{ is } g\text{-dangling}\} \\
&= \binom{|V_p|}{g} \mathbb{P}(T(S)) \, \mathbb{P}(O(S)) \, \mathbb{P}(C(S)|O(S), T(S)) \\
&= \binom{|V_p|}{g} \mathbb{P}(|E(S)| = g - 1) \mathbb{P}\left(T(S) \big| |E(S)| = g - 1\right) \, \mathbb{P}(O(S)) \, \mathbb{P}(C(S)|O(S), T(S))
\end{aligned}
$$

Each term is bounded from below as follows. Let $\lfloor x \rfloor$ denote the largest integer less than $x$. Then there exists $\epsilon_1 > 0$, such that

$$
\binom{|V_p|}{g} > \binom{\lfloor \eta N \rfloor}{g} > \frac{\eta^g N^g}{g^g} > \epsilon_1 N^g.
$$

Then, because edge probabilities are bounded between $1/N$ and $b/N$,

$$
\mathbb{P}(|E(S)| = g - 1) > \binom{\binom{g}{2}}{g - 1} (1/N)^{g-1} (1 - b/N)^{\binom{g}{2} - (g-1)} > \epsilon_2 N^{-(g-1)}.
$$

The next probability is the probability that the $g - 1$ edges in $S$ form a tree. This does not depend on $N$.

$$
\mathbb{P}\left(\text{the node induced graph from } S \text{ is a tree} \big| |E(S)| = g - 1\right) > \epsilon_3
$$

Then, $O(S)$ is bounded similarly to $|E(S)|$.

$$
\mathbb{P}(O(S)) > \binom{(N - g)g}{1} (1/N)^1 (1 - b/N)^{\binom{(N-g)g}{1} - 1} > \epsilon_4 (N - g)/N > \epsilon_5
$$

Given $O(S)$ and $T(S)$, the condition $C(S)$ is certainly satisfied if the one external edge connects to a component that is larger than $\epsilon_6 N$. Because we are only considering models with $p_{ij} > (1 + \epsilon)/N$, these graphs are all more connected than an Erdös-Rényi graph with $p = (1 + \epsilon)/N$. Even after removing the set $S$, the size of the largest connected component of such an Erdös-Rényi graph is greater than $\epsilon_6 N$ a.s. [1]. As such $\mathbb{P}(C(S)|O(S), T(S)) > \epsilon_7$.

Putting the bounds together,

$$
\mathbb{E}D_g > \epsilon_8 N^g N^{-(g-1)} = \epsilon_8 N.
$$

This concludes the proof. $\qquad \square$

We must still prove Lemma 1.2. The proof of Lemma 1.2 requires the next fact.

**Fact 1.3.** *For any $g$-dangling set, there is one edge connecting the $g$-dangling set to its connected component in $G$. If that edge is removed, then there are two connected components: the graph on the $g$-dangling set and a larger graph of at least $9g$ nodes.*

Here is a proof of Lemma 1.2.

*Proof.* Suppose that $i$ is contained in two $g$-dangling sets, $S \neq \tilde{S}$. Because they are not equal, there must be a node $q$ such that $q \in \tilde{S}$ and $q \notin S$. Because $\tilde{S}$ is $g$-dangling, $i$ must have unique path to $q$

that falls within $\tilde{S}$. Because $q$ is outside of $S$, that unique path must include the unique "bridge edge" that connects $S$ to $S^c$. Define that bridge edge to be $(b, k)$. This implies that $b, k \in \tilde{S}$.

There must also be a node $\ell$ such that $\ell \in S$ and $\ell \notin \tilde{S}$. For any node $j \in (S \cup \tilde{S})^c$, every path from $\ell$ to $j$ must include the bridge edge $(b, k)$ for $S$. Because $\ell \notin \tilde{S}$, the unique path within $S$ from $i$ to $\ell$ must contain the bridge edge for $\tilde{S}$. Now, consider dropping the bridge edge for $\tilde{S}$. From Fact 1.3, this must create 2 connected components, one of size $g$ and another greater than $9g$. That large component must contain both $\ell$ and the nodes in $(S \cup \tilde{S})^c$. This leads to a contradiction because $\ell$ cannot be path connected to $(S \cup \tilde{S})^c$ without the edges in $\tilde{S}$. $\qquad\square$

## 2 Proof of Theorem 3.5

*Proof.* Note that because this theorem only discusses unweighted graphs, $w_{ij}$ is either zero or one.

Denote the $Q$ $g$-dangling sets as $\{S_{\epsilon_l}\}_{l=1}^{Q}$. For each $g$-dangling set $S_{\epsilon_l}$, we define its cluster identifier as $f^{(l)} = (f_1^{(l)}, \ldots, f_N^{(l)}) \in \mathbb{R}^N$, where each element is

$$f_i^{(l)} = \begin{cases} \sqrt{\frac{d_i}{vol(S_{\epsilon_l})}} & \text{if } i \in S_{\epsilon_l} \\ 0 & \text{otherwise} \end{cases}.$$

Then,

$$\begin{aligned} f^{(l)T} L f^{(l)} &= \frac{1}{2} \sum_{i \sim j} w_{ij} \left( \frac{f_i^{(l)}}{\sqrt{d_i}} - \frac{f_j^{(l)}}{\sqrt{d_j}} \right)^2 \\ &= \sum_{i \sim j, i \in S_{\epsilon_l}, j \notin S_{\epsilon_l}} \frac{w_{ij}}{vol(S_{\epsilon_l})} \\ &= \frac{1}{vol(S_{\epsilon_l})}, \end{aligned}$$

The first equality is from Prop 3 in [3]. The last equality is because there is only one edge connecting $S_{\epsilon_l}$ and $S_{\epsilon_l}^c$.

Each $g$-dangling set $S_{\epsilon_l}$ has an identifier $f^{(l)}$. Similarly, we define an identifier $f^{(0)} \in \mathbb{R}^N$ for the set $S_0 = (\cup_{l=1}^{Q} S_{\epsilon_l})^c$, where each element is

$$f_i^{(0)} = \begin{cases} \sqrt{\frac{d_i}{vol(S_0)}} & \text{if } i \in S_0 = (\cup_{l=1}^{Q} S_{\epsilon_l})^c \\ 0 & \text{otherwise} \end{cases}.$$

Then,

$$\begin{aligned} f^{(0)T} L f^{(0)} &= \frac{1}{vol(S_0)} \sum_{i \sim j, i \in S_0, j \notin S_0} w_{ij} \\ &\leq \frac{Q}{vol(S_0)}, \end{aligned}$$

The inequality is because there are at most $Q$ edges connecting $S_0$ with $\cup_{l=1}^{Q} S_{\epsilon_l}$ (one for each $g$-dangling set).

Thus, sum of the leading $Q + 1$ eigenvalues of $L$

$$\sum_{i=0}^{Q} \lambda_i = \min_{V^T V = I_{Q+1}} trace(V^T L V)$$

$$\leq f^{(0)T} L f^{(0)} + \sum_{l=1}^{Q} f^{(l)T} L f^{(l)}$$

$$\leq \frac{Q}{vol(S_0)} + \sum_{l=1}^{Q} \frac{1}{2g-1}$$

$$= Q \left( \frac{1}{vol(S_0)} + \frac{1}{2g-1} \right)$$

$$< \frac{Q}{2g-2}.$$

The first equality is from Ky Fan Maximum Principal [2]. The last inequality is from the condition $vol(S_0) \geq 4g^2$.

Thus, at least $Q/2$ eigenvalues are no larger than $(g-1)^{-1}$. $\qquad\square$

## 3 Proof of Corollary 4.7

**Proof of Proposition 4.5**

*Proof.*

$$\texttt{CoreCut}_\tau(S_\epsilon) = \frac{cut(S_\epsilon, G) + \frac{\tau}{N} |S_\epsilon||S_\epsilon^c|}{vol(S_\epsilon, G) + \tau |S_\epsilon|} \geq \frac{\frac{\tau}{N} |S_\epsilon^c|}{\bar{d}(S, G) + \tau} \geq \frac{|S_\epsilon^c|}{N} \frac{\alpha}{1+\alpha} > \frac{\alpha(1-\epsilon)}{1+\alpha}$$

The first inequality is by dividing $|S_\epsilon|$ in both numerator and denominator. The second inequality is from the assumption $\tau \geq \alpha \bar{d}(S_\epsilon, G)$. The last inequality is from assumption $|S_\epsilon| < \epsilon N$. $\qquad\square$

**Proof of Proposition 4.6**

*Proof.*

$$\texttt{CoreCut}_\tau(S) = \frac{cut(S, G) + \frac{\tau}{N} |S||S^c|}{vol(S, G) + \tau |S|} = \frac{\phi(S, G) + \tau |S^c|/(N\bar{d}(S, G))}{1 + \tau/\bar{d}(S, G)}$$

$$< \frac{\phi(S, G) + \tau/\bar{d}(S, G)}{1 + \tau/\bar{d}(S, G)} < \phi(S, G) + \delta$$

The second equality is by dividing $vol(S)$ in both numerator and denominator. The first inequality is from $|S^c| < N$. The second inequality is from $\tau/\bar{d}(S, G) \in (0, \delta)$. $\qquad\square$

Corollary 4.7 follows directly from Proposition 4.5 and Proposition 4.6.

## 4 More simulations and data examples

Figure 1 compares the leading eigenvectors of `Vanilla-SC` and `Regularized-SC` on the referral network among Wisconsin primary physicians based on 2013 Medicare provider utilization and payment data. Figure 2 compares the leading eigenvectors using the brain graph from `https://neurodata.io`.

Figure 1: The leading eigenvectors of the Wisconsin physician referral network.

Figure 2: The leading eigenvectors of the brain graphs. The left three columns contain eigenvectors for `Vanilla-SC` and the right three columns contain eigenvectors for `Regularized-SC`.