[Reviews · NeurIPS 2018]

Reviewer 1



The paper provides a graph theoretic connection to regularized spectral clustering for community detection in networks. The spectral clustering method of community detection can also be interpreted as relaxation of normalized cut measure of subgraphs of networks. The paper proves that regularized spectral clustering can be seen as a relaxation of similar cut measure of subgraphs, which has been called Core-cut in the paper. The paper provides theoretical proof on connection between regularized spectral clustering and Core-cut measure. The intuition of Core-cut measure is given for sparse inhomogeneous graphs and its representation in terms of core-periphery model. The paper also provides properties of Core-cut measure, which are similar to normalized cut measures. However, the paper states 'Facts' in the paper and supplementary and it is not clear what kind of statements are the 'Facts'. It would be better to either give a proof of the 'Facts' or refer to paper/books where the proof can be found. The paper is generally well-written and gives adequate intuition for the results and definitions provided in the paper. The paper considers recently developed regularizations in spectral clustering for sparse graphs. To my knowledge, this is new work and nice attempt to link cut measures of graph to regularization and try to find significance of regularization by analyzing the Core-cut measure. The paper is of more theoretical in nature. It mostly provides a connection between cut measures in graphs and regularization in spectral clustering. Thus, this paper gives a strong case for regularization in spectral clustering, by showing that the Core-cut measure is better in partitioning networks with core-periphery structure often seen in sparse inhomogeneous graphs.

Reviewer 2



I am familiar with spectral clustering and its relation to graph conductance, but do not know much about the references on regularized SC, such as [17, 9, 11]. So my assessment on regularized SC is based on what I learnt from this work. Here are my comments: + The paper provides a good understanding of regularized-SC for sparse-graph partitioning based on solid theoretical reasoning. The extensive experimental results also well justify the statements. As a large fraction of social networks are sparse, so the assumptions are acceptable. + The paper is written well and easy to follow. All theoretical results are presented well and show insight on why regularized-SC is important (Thm 4.3) and when spectral SC can work well(Corollary 4.7) - The main concern is about the usage of regularized-SC in practice. I agree that the when using spectral clustering without regularization, the first-round partition may yield a huge component and several small parts like the dangling sets mentioned in this work. However, in practice, we can easily do another round of graph partition by focusing on the obtained huge component to refine the clustering results. Regularized-SC may not require such additional effort but it introduces a parameter \tau whose tuning is not easy. Two suggestions: -The format of 2-way-partition regularized-SC seems equivalent to global Pagerank where \tau is to control the probability of random jump. Pagerank algorithm is widely used everywhere. It would better to give some discussion on this point and review related references. Some work on Pagerank also shows connection with graph conductance such as Andersen et al. Local graph partitioning using pagerank vectors [FOCS2006] - Since sparsity comes from the real social networks, it may be better to provide some real evidence on the number of dangling sets in social networks. Moreover, as shown in the appendix, Thm 3.4 requires another assumption which should be added within the statement in the main text. Some minor typo: Line 29: sigular -> singular Line 31: tau -> \tau Line 75:covolutional -> convolutional

Reviewer 3



The focus of the paper is on spectral clustering, and more specifically on its tendency to detect many small clusters instead of the actual structure of the graph. This property is first quantified using some random graph model. It is then shown how edge-wise regularization can be used to find relevant clusters. Some experiments on real datasets illustrate the effectiveness of the approach. The paper is interesting, topical and well written. The main criticism I have is that only simple cuts are considered. The extension of the results to more than 2 clusters is not even mentioned. The gain in computation times shown in Figure 7 should also be discussed: why does rARPACK work faster on the regularized Laplacian?